# Habitat Imaging Biomarkers for Diagnosis and Prognosis in Cancer Patients Infected with COVID-19

**DOI:** 10.3390/cancers15010275

**Published:** 2022-12-31

**Authors:** Muhammad Aminu, Divya Yadav, Lingzhi Hong, Elliana Young, Paul Edelkamp, Maliazurina Saad, Morteza Salehjahromi, Pingjun Chen, Sheeba J. Sujit, Melissa M. Chen, Bradley Sabloff, Gregory Gladish, Patricia M. de Groot, Myrna C. B. Godoy, Tina Cascone, Natalie I. Vokes, Jianjun Zhang, Kristy K. Brock, Naval Daver, Scott E. Woodman, Hussein A. Tawbi, Ajay Sheshadri, J. Jack Lee, David Jaffray, Carol C. Wu, Caroline Chung, Jia Wu

**Affiliations:** 1Department of Imaging Physics, MD Anderson Cancer Center, 1515 Holcombe Blvd, Houston, TX 77030, USA; 2Department of Radiation Oncology, MD Anderson Cancer Center, Houston, TX 77054, USA; 3Department of Enterprise Data Engineering & Analytics, MD Anderson Cancer Center, Houston, TX 77054, USA; 4Department of Neuroradiology, MD Anderson Cancer Center, Houston, TX 77054, USA; 5Department of Thoracic Imaging, The University of Texas MD Anderson Cancer Center, Houston, TX 77054, USA; 6Department of Thoracic/Head and Neck Medical Oncology, MD Anderson Cancer Center, Houston, TX 77054, USA; 7Department of Genomic Medicine, MD Anderson Cancer Center, Houston, TX 77054, USA; 8Department of Leukemia, MD Anderson Cancer Center, Houston, TX 77054, USA; 9Department of Melanoma Medical Oncology, MD Anderson Cancer Center, Houston, TX 77054, USA; 10Department of Pulmonary Medicine, MD Anderson Cancer Center, Houston, TX 77054, USA; 11Department of Biostatistics, The University of Texas MD Anderson Cancer Center, Houston, TX 77054, USA; 12Office of the Chief Technology and Digital Officer, MD Anderson Cancer Center, Houston, TX 77054, USA; 13Office of the Chief Data Officer, MD Anderson Cancer Center, Houston, TX 77054, USA

**Keywords:** COVID-19, habitat imaging, machine learning, diagnosis, prognosis

## Abstract

**Simple Summary:**

Patients with cancer are often immuno-compromised, and at a high risk of experiencing various COVID-19-associated complications compared to the general population. Additionally, COVID-19 infection and lung toxicities due to cancer treatments can present with similar radiologic abnormalities, such as ground glass opacities or patchy consolidation, which poses further challenges for developing AI algorithms. To fill the gap, we carried out the first imaging AI study to investigate the performance of habitat imaging technique for COVID-19 severity prediction and detection specifically in the cancer patient population, and further tested its performance in the general population based on multicenter datasets. The proposed COVID-19 habitat imaging models trained separately on the cancer cohort outperformed those AI models (including deep learning) trained on the multicenter general population by a significant margin. This suggests that publicly available COVID-19 AI models developed for the general population will not be optimally applied to cancer.

**Abstract:**

Objectives: Cancer patients have worse outcomes from the COVID-19 infection and greater need for ventilator support and elevated mortality rates than the general population. However, previous artificial intelligence (AI) studies focused on patients without cancer to develop diagnosis and severity prediction models. Little is known about how the AI models perform in cancer patients. In this study, we aim to develop a computational framework for COVID-19 diagnosis and severity prediction particularly in a cancer population and further compare it head-to-head to a general population. Methods: We have enrolled multi-center international cohorts with 531 CT scans from 502 general patients and 420 CT scans from 414 cancer patients. In particular, the habitat imaging pipeline was developed to quantify the complex infection patterns by partitioning the whole lung regions into phenotypically different subregions. Subsequently, various machine learning models nested with feature selection were built for COVID-19 detection and severity prediction. Results: These models showed almost perfect performance in COVID-19 infection diagnosis and predicting its severity during cross validation. Our analysis revealed that models built separately on the cancer population performed significantly better than those built on the general population and locked to test on the cancer population. This may be because of the significant difference among the habitat features across the two different cohorts. Conclusions: Taken together, our habitat imaging analysis as a proof-of-concept study has highlighted the unique radiologic features of cancer patients and demonstrated effectiveness of CT-based machine learning model in informing COVID-19 management in the cancer population.

## 1. Introduction

The global pandemic of the COVID-19 disease forced researchers to swiftly develop effective ways to mitigate the quick spread of the virus. Accurate diagnostic solutions have been developed for COVID-19, among which Reverse Transcription Polymerase Chain Reaction (RT-PCR) is considered as the gold standard in detecting the infection. However, RT-PCR has several limitations with a false-negative rate [1,2,3] that is sufficiently high and multiple tests may be required to confirm the diagnosis of SARS-CoV-2 infection. More importantly, RT-PCR offers no insight on assessing disease severity to guide patient management. Due to these drawbacks, several approaches were investigated to complement the use of the RT-PCR, including clinical symptoms, laboratory findings, and imaging. For imaging, computed tomography (CT) and chest radiography (CXR) are routinely used to guide clinicians in the diagnosis and assess the pulmonary severity of COVID-19 [4,5,6,7,8].

Patients with cancer are often immuno-compromised and at a high risk of experiencing various COVID-19 associated complications compared to the general population [9]. Pilot artificial intelligence (AI) algorithms largely focus on the general population, utilizing features such as clinical symptoms [10,11], lab tests [12,13,14], and CT findings [15,16,17] to diagnose COVID-19 and predict outcome after infection. While these AI algorithms showed initial promise, there were challenges and failures observed in the clinical implementation of these models driven by a failure to validate algorithms in heterogeneous populations as well as a change in disease presentation with SARS-CoV-2 variants [18]. The likelihood for the clinical utility of models built using data from the general population is even smaller in immunocompromised hosts, who may bear distinct risk factors for severe disease. Additionally, COVID-19 infection and lung toxicities due to cancer treatments can present with similar radiologic abnormalities, such as ground glass opacities (GGO) or patchy consolidation [19,20,21] on chest CT, which poses further challenges for AI detection algorithms. Though pilot AI models have been developed to predict COVID-19 severity and deterioration specifically in cancer patients [22,23], they left out the quantitative imaging metrics. Further, it is unclear whether and to what degree these AI models built on the cancer population are different from the ones on the general population, which is what we aim to investigate in this study.

Imaging findings in COVID-19 vary greatly from patient to patient, including in the extent and heterogeneity of involvement and the characteristics of the lung infiltrates. Radiologists score pneumonia severity by assessing the percentage and distribution of distinct infected regions, such as ground-glass opacities (GGO) and consolidation. These standards are known to be coarse, subjective, and not robust enough to characterize complex infection patterns. Many AI approaches (radiomics and deep learning) attempt to automate these measurements through crafting robust computational pipelines. On the other hand, habitat imaging offers an avenue for us to define intrinsic infection patterns. We have shown that habitat imaging quantifies intratumoral heterogeneity in cancer patients [24]. Unlike classical radiomics analysis which treats the heterogeneous tumor as one entity, habitat imaging explicitly partitions the complex tumor into phenotypically distinct subregions, where these intratumoral subregions are termed habitats. These subregions have differing prognostic implications for cancer severity. Similar to genomic sequencing studies that show the clonal diversity of cancer cells [25], habitat imaging offers a new and powerful avenue to investigate how molecular diversity manifests on the radiological scans, and we have demonstrated added value in predicting treatment response [26].

The main goal of this study is to investigate the performance of habitat imaging technique for COVID-19 prognosis (primary aim) as well as detection (secondary aim) in cancer patient population and compare its performance to the general population. Different from existing AI studies that mainly focused on the general population, here we specifically focus on the COVID-vulnerable cancer population.

## 2. Materials and Methods

### 2.1. Overall Study Design

The overall goal is to develop and test imaging biomarkers for the diagnosis and prognosis of COVID-19 in the cancer population and further compare its performance in the general population (dataset details in Table 1). To achieve this goal, we proposed a habitat imaging framework that consists of three main steps (Figure 1). First, we applied imaging preprocessing and fusion pipeline to highlight the lung infection patterns. Second, an unsupervised subregion segmentation approach (i.e., habitat analysis) is proposed to reveal these complex infection patterns by partitioning the whole lung regions into phenotypically different habitats. Third, we characterized the spatial arrangement/interaction among these habitats and built machine learning models for COVID-19 detection and severity prediction.

### 2.2. Patient Population

This retrospective study was approved by the MD Anderson institutional review board and compliant with the Health Insurance Portability and Accountability Act. We enrolled two types of patient population, including a general population set and a cancer population set (Table 1).

**Table 1 cancers-15-00275-t001:** Demographic and clinical characteristics of general and cancer population. Note, There are three datasets, Stony Brook (N = 275 patients with 304 CT scans), RICORD COVID-19 positive (N = 110 patients with 110 CT scans) and RICORD COVID-19 negative (N = 117 cases with 117 CT scans) that we referred to as the general population. Similarly, we have three datasets, MD Anderson COVID-19 positive (N = 252 patients with 258 CT scans), Leukemia (N = 126 patients with 126 CT scans) and Melanoma (N = 36 patients with 36 CT scans) that we referred to as cancer population. The Leukemia and Melanoma subcohorts are not highlighted in this table as the detailed demographic information for these datasets is not available.

Datasets Characteristics	Stony Brook (N = 275)	MD Anderson COVID-19 Positive (N = 252)
	No. (%)	No. (%)
**Age (years)**		
<18	-	1 (0.4%)
18–59	125 (45%)	112 (44%)
60–74	83 (30%)	99 (39%)
75–90	67 (24)	37 (15%)
>90	-	3 (1%)
**Sex**		
Male	160 (58%)	121 (48%)
Female	106 (39%)	131 (52%)
NA	9 (3%)	-
**BMI**		
>30	100 (36%)	70 (28%)
<30	127 (46%)	95 (38%)
NA	48 (17%)	87 (35%)
**Smoking status**		
Current	6 (2%)	11 (5%)
Former	63 (23%)	90 (36%)
Never	138 (50%)	148 (59%)
NA	68 (25%)	3 (1%)
**Major diseases**		
Malignancy	25 (9%)	240 (95%)
Hypertension	105 (38%)	136 (54%)
Diabetes	54 (20%)	105 (42%)
Coronary artery diseases	33 (12%)	50 (20%)
Chronic kidney disease	19 (7%)	75 (30%)
Chronic obstructive pulmonary disease	18 (7%)	15 (6%)
Other lung diseases	39 (14%)	22 (9%)
**Symptoms at onset**		
Fever	158 (57%)	33 (13%)
Shortness of breath	152 (55%)	14 (6%)
Cough	157 (57%)	39 (15%)
Nausea	49 (18%)	17 (7%)
Vomiting	29 (11%)	9 (4%)
Diarrhea	65 (24%)	17 (7%)
Abdominal pain	19 (7%)	7 (3%)
**Hospitalization status**		
Inpatient/admitted	259 (94%)	86 (34%)
Outpatient	-	166 (66%)
Emergency Visit	16 (6%)	-
**ICU**		
TRUE	83 (30%)	42 (17%)
FALSE	192 (70%)	209 (83%)
**Oxygen requirement**		
TRUE	71 (26%)	20 (8%)
FALSE	204 (74%)	232 (92%)

For the general population, a total of 531 CT scan images were obtained from public TCIA database (Stony Brook [27,28] and RICORD [28,29,30] cohorts), including 385 patients with confirmed COVID-19 positive with RT-PCR and 117 COVID-19 negative patients from four international sites. For the cancer population, a total of 420 CT scan images from 414 cancer patients receiving treatment at MD Anderson cancer center were included. Among the 414 cancer patients, 252 patients were confirmed to have COVID-19 infection with RT-PCR while the remaining 162 patients were COVID-19 negative per RT-PCR but had other causes of pneumonia or pneumonitis. We utilized the available CT scan images of same patient at multiple timepoints to build the diagnostic models. For the prognostic model, only the CT scans closest to the PCR test were used to predict the end points (hospitalization, ICU admission and ventilation).

### 2.3. Imaging Preprocessing and Fusion

We utilized the U-net [31] to segment the right and the left lung from the original CT images. Then, the CT density of lungs was normalized using both the lung and mediastinum windows. Since the infected regions within the lungs manifested as ground glass opacity (GGO), consolidation, and their mixture, which usually corresponded to increased heterogeneity compared to the normal lung parenchyma, we specifically characterized these local density and texture variations using the entropy filter. For a small neighborhood $N_{l}$ with window size $n_{l}\times m_{l}$ within an image I, we can compute the local entropy of the neighborhood $N_{l}$ as:(1)$$E\left(N_{l}\;\right)=-\sum_{r=0}^{M-1}p_{r}\log p_{r}$$
where $p_{r}=q_{r}/\left(n_{l}\times m_{l}\right)$ denotes the probability of the grayscale r appearing in the neighborhood $N_{l}$ and $q_{r}$ denotes the number of pixels with grayscale r in $N_{l}$.  M is the maximum grayscale. The filtered image channels were fused together to form the final image on which the habitat analysis is applied. Figure 1 shows an example of the lung window and mediastinum window, the corresponding results of their filtered local entropy maps (Equation (1)), and the fused image. Note that brighter pixels in the filtered maps correspond to neighborhoods with high local entropy in the original images.

### 2.4. Habitat Image Pipeline

After imaging harmonization, the habitat imaging is an unsupervised (clustering based) approach [26,32,33] containing two clustering modules at both individual and global levels, as illustrated in Step 2 of Figure 1. The individual level module is designed to segregate each patient’s lungs into superpixels by grouping neighboring pixels of similar intensity and texture, whereas the global level module was designed to refine the individual-level over-segmentation by exploiting the similarity of both inter and intra-patient superpixels.

Individual level clustering module

Here, we aimed to partition each of the input 3D lungs into phenotypically similar small pieces. Given the fused maps, the simple linear iterative clustering (SLIC) [34] algorithm was utilized to over-segment them into a cluster of superpixels. The main idea behind the SLIC superpixel algorithm is to adopt the local kmeans algorithm to generate clusters grouped by neighboring voxels of similar imaging patterns. In detail, the SLIC algorithm involves two steps:

**Step 1**. Cluster center initialization: The algorithm starts by splitting the input image into K regular grids and then chooses the center of each grid as the initial cluster centers. To avoid assigning a cluster center on image edge, the edge pixels with the lowest gradient in a 3×3×3 neighborhood is chosen as the initial seed location.

**Step 2**. Local kmeans clustering: Each pixel is then assigned to the nearest cluster center using a distance measure D:(2)$$D\left(i,j\right)=\sqrt{{\left(\frac{d_{c}\left(i,j\right)}{m}\right)}^{2}+{\left(\frac{d_{s}\left(i,j\right)}{S}\right)}^{2}}$$
where $d_{c}\left(i,j\right)=\left|a_{i}-a_{j}\right|$ and $d_{s}\left(i,j\right)=\sqrt{{\left(x_{j}-x_{i}\right)}^{2}+{\left(y_{j}-y_{i}\right)}^{2}+{\left(z_{j}-z_{i}\right)}^{2}}$, $a_{i}$ denotes the mean density of the voxels, $d_{c}$ denotes the density proximity, $d_{s}$ denotes the spatial proximity, S denotes the initial grid step (sampling interval of the cluster centroids) and m is a parameter that controls the compactness of the superpixels. The distance (Equation (2)) is computed within a confined region around the cluster center and this step is repeated until the maximum number of iterations is reached or the residual error converges.

Global level clustering module

We merged the superpixels obtained for individual patients to study the inter and intra patient similarity, so that superpixels with similar features/characteristics within a lung were fused to form a habitat (subregion). Moreover, corresponding subregions across the entire population were consistently labeled. In particular, each superpixel was represented by first order statistics of its encompassing voxels’ four channels (lung and mediastinum windows, and their entropy), and subsequently, kmeans clustering algorithm was used to define the optimal habitat regions.

Multiregional spatial interaction (MSI) feature extraction

The multiregional spatial interaction (MSI) matrix was used to quantify the intra-lung infection heterogeneity on the habitat maps (Step 3 of Figure 1). In detail, for any voxel within each lung, we scanned its neighbor and the co-occurring pairs were added to the corresponding cell in the MSI matrix (Step 3 of Figure 1). After looping through all lung voxels, the habitats’ spatial distribution and interaction patterns were abstracted in this MSI matrix. Subsequently, a set of quantitative features were extracted from the MSI matrix including first order and second order statistical features. The first order statistical features include the absolute volume and relative proportion of individual subregions as well as their interacting boarders, whereas the second order statistical features summarize the subregion’s spatial heterogeneity, including the contrast, homogeneity, correlation, and energy.

### 2.5. Habitat Imaging Based Machine Learning Model

To improve their generalizability, we aimed to build a parsimonious model by selecting features with high discriminant power. Two clinical endpoints were studied: (1) COVID-19 diagnosis; (2) prognosis, including admission type: outpatient vs. inpatient, ICU, and ventilation. Given these extracted features, we used a univariate feature ranking approach (chi-square test) to evaluate each feature’s association with individual endpoints. The feature importance score is computed based on the p-value of univariate testing (−log(p)). The features were ranked by computed scores where feature with the largest score was considered as the most important feature. We then iteratively selected the top ranked features to build four types of machine learning models, including logistic regression, generalized additive model, support vector machine, and random forest. 10-fold cross validation was used for parameter tuning and performance evaluation. Of note, the datasets for COVID-19 prognosis and severity are highly imbalanced (see Table 1). To avoid evaluation bias (e.g., assigning all cases to the dominant class), we employed the oversampling approach to augment samples from the underrepresented classes to balance their cases with majority class.

### 2.6. Model Interpretation

To understand the constructed habitat models, we systematically investigated its meaning. First, we overlapped the habitat subregions with radiologist’s manual annotation of COVID infected regions on CT. Then, we correlated the MSI features to the radiologist’s semantic readings. In particular, the radiologists, blinded from habitat modeling, reported CT features in structured report designed based on the RSNA COVID-19 reporting template (https://radreport.org/home/50830/2020-07-08%2011:50:07 accessed on 1 November 2021). The semantic features include presence of consolidation and/or ground glass opacities (GGO), and if present, the laterality, location, quantity GGO/consolidation, and patterns/morphology of GGO, In addition, presence or absence of centrilobular nodule, discrete solid nodule, lymphadenopathy, bronchial wall thickening, mucoid impaction, pericardial effusion, pleural effusion, pulmonary embolism, smooth septal thickening, endotracheal tube, pulmonary cavities were also noted. COVID classification patterns based on the RSNA Consensus Statement [35] and CT severity score were also recorded. Volcano plot was utilized to visualize the association of the MSI features with individual semantic readings.

### 2.7. Statistical Analysis

The receiver operating characteristics (ROC) curve analysis as well as the area under the curve (AUC) were used to evaluate the prediction capability of the habitat imaging models. The optimal threshold to separate different classes was defined based on the Youden’s J statistics during training and the same threshold was applied during validation. We also reported the model’s sensitivity, specificity, and accuracy. These metrics were evaluated using 10-fold cross validation scheme.

## 3. Results

### 3.1. Habitat Image Analysis

We independently applied the habitat pipeline in both the general and cancer cohorts, and consistently identified six distinct habitats (subregions) at the population level, corroborating prior study of COVID-19 infection [36]. Figure 2 shows the detailed distribution of four input image channels across the six habitat regions as well as their imaging interpretations. In general, these habitat subregions are phenotypically different and that the individual subregions were associated with distinct imaging features. The subregion 4 corresponds to the background normal lung parenchyma, and subregion 2 corresponds to the pulmonary effusion. Subregion 3 corresponds to the dense and homogeneous consolidated area, and subregion 1 corresponds to pure GGO. Interestingly, subregions 5 and 6 correspond to the infected areas from GGO to consolidation at different degrees.

Clinically, it remains a laborious task to manually delineate areas of involvement on each CT image, especially given the heterogeneous infection patterns. We compared the habitat maps with annotation of lung infection by radiologists. As shown in Appendix A, the infected regions segmented by the radiologist were mostly captured and separated from the normal lung parenchyma region using proposed habitat approach. Furthermore, the habitat imaging analysis can partition these heterogeneous infected regions into six COVID-habitat regions (Figure 3).

### 3.2. Habitat Models for COVID-19 Diagnosis

Given the habitat maps, fifty-eight multiregional spatial interaction features (details in Table 2) were measured from these subregions for COVID-19 detection. We conducted both unsupervised and supervised machine learning analysis to examine the diagnostic values of habitat features for COVID-19 detection.

First, we applied low-rank embedding algorithms to assess the performance of habitat features for differentiating COVID-19 positive from COVID-19 negative cases when mixing the general and cancer patients (Figure 4). The COVID-19 positive cases from the general population (RICORD, Stony Brook) and cancer population (MD Anderson Population) tend to mix and form a tight cluster, indicating high similarity among them as measured by habitat analysis. Since these embedding algorithms are designed to project the inherent local structure of high dimensional data to low dimensional subspace, the clear separation of COVID-19-positive from negative cases indicates that the habitat-derived MSI features can effectively encode discriminant imaging information. Interestingly, the MSI features can also differentiate the non-COVID-associated pneumonia across different cancers, including pneumonia in acute myeloid leukemia or melanoma cases.

Next, we built separate prediction models to detect COVID-19 infection on the general population and cancer population (Figure 5 and Table 3). For the general population, the optimized LR, SVM, RF and GAM models were fitted with the top 23, 10, 13, and 7 features, respectively, and their corresponding confusion matrices were presented in Figure 5A. Similarly, for the cancer population, the performances of these models using different number of features were reported in the accuracy and AUC curves (Figure 5B), where these models achieved high performance for COVID-19 detection.

We compared the performance of our proposed habitat imaging approach to several reported COVID-19 detection approaches from either conventional radiomics or deep learning approaches based on imaging or other data source. As can be seen in Appendix A, our habitat imaging approach significantly outperformed the existing approaches.

### 3.3. Comparison between Diagnostic Models of General Population and Cancer Population

We first examined the feature importance ranked in two models for general or cancer population (see Figure 5) for COVID-19 diagnosis. The top ranked features in the general cohort are significantly different from cancer cohort. For instance, MSI25 which captures interaction between SR2 and SR6 appears to be the most important predictor in the general cohort, while it is not top ranked in cancer. The fluctuation across the two different cohorts is possibly due to their underlying difference in pneumonia patterns as appeared on CT scans.

Next, we tested the performance of machine learning model trained on general population directly on cancer population, where we observed a decreased prediction capacity compare with the dedicated cancer model as shown in Appendix A. In addition, when finetuning the general population model on the cancer population, we observed that the mixture of two heterogenous populations adversely affected model’s performance (Table 4).

### 3.4. Habitat Models for COVID-19 Prognosis and Severity Prediction

We tested the effectiveness of habitat imaging models in predicting COVID-19 disease severity with three end points include patient’s admission, need for intensive care, and need of mechanical ventilation. The habitat imaging-based classification models achieved high performance for the prognosis analysis, shown in Table 5, Appendix A, and Figure 6 and Appendix A. The prognostic models were able to obtain an AUC score of 1 for ICU and ventilation prediction on the cancer population, and AUC scores ranged from 0.96 to 1 on general population. In addition, we found the optimal prediction of the RF and GAM models using fewer number of selected features than LR and SVM models, where the curves started to plateau early on a handful of features. By contrast, the LR and SVM models kept improving with the increase in number of selected features, where they reached optimal performance using almost all the features. For example, the LR and SVM models achieved the best accuracy and AUC scores for admission prediction on cancer cohort using a total of 49 and 51 top-ranked features, respectively. Taken together, the habitat-based machine learning models can accurately infer the severity of a COVID-19 infection in both general and cancer population, by predicting whether the infected patients need to be admitted to hospital and ICU and also if they require ventilation.

### 3.5. Habitat Models Interpretation

We performed the correlation analysis between habitat imaging features and semantic readings from radiologists. A limited number of habitat features were observed to associate with radiologist’s readings and reached the predefined statistical significance level at a false discovery rate (FDR) of <0.05 as shown in Appendix A. For instance, MSI17 that measures interaction of habitat regions 1 and 2 and MSI19 that measures interaction of habitat regions 1 and 3 were correlated to the presence of ground glass opacities (GGO). Additionally, MSI17, that measures volume of subregion 3, was negatively correlated to presentation of pulmonary embolism, and it was found to be correlated to quantity of GGO consolidation and solid nodules.

### 3.6. Comparison to Deep Learning Approach

We further examine the performance of deep learning model for COVID-19 diagnosis and prognosis in the general and cancer population. Specifically, we trained a DenseNet121 model using the stochastic gradient descent (SGD) method with momentum of 0.9 and initial learning rate of 0.001. To avoid fluctuation in the later stage of training, we set the learning rate to decay by 0.9 every 10 epochs. We consider the binary cross entropy as the loss function. Inputting the CT scan images, we extracted deep features (N = 1024) at the last layer of the model and used the extracted deep features to train the same classifiers as in the habitat analysis approach.

For COVID-19 diagnosis in the general and cancer population, the best performances in terms of AUC were obtained using the LR and RF models with AUC’s 0.9792 and 0.9750, respectively (Appendix A). When training the different classification models on the general population and finetuning on the cancer population, the SVM model obtained the best AUC result of 0.9673 (Appendix A). The SVM model performed the best for admission, ventilation and ICU prediction in the cancer population with AUC values of 0.9924, 1.0000 and 1.0000, respectively (Appendix A). Similar performance of the SVM model is observed in the general population with AUC value of 1.0000 for both admission and ventilation prediction, while GAM obtained the best AUC value of 0.9864 for ICU prediction (Appendix A).

## 4. Discussion

In this study, we developed and validated the habitat imaging approach for COVID-19 severity prediction and detection using chest CT of a cancer population instead of a general population. Our proposed habitat analysis has demonstrated robust performance in partitioning the whole infected lungs into phenotypically distinct subregions. By quantifying the spatial distributions and interactions of these intrinsic subregions using the hand-crafted multi-regional spatial interaction features, machine learning models were built and showed high performance in COVID-19 infection diagnosis. More importantly, the habitat models demonstrated high accuracy for predicting infection severity, including the needs for hospital admission, ICU stay, and ventilation support. Taken together, our habitat imaging analysis as a proof-of-concept study has demonstrated the effectiveness of CT-based machine learning model in informing COVID-19 management in the cancer population, if prospectively validated.

To the best of our knowledge, this is the first reported imaging study that explores the performance of machine learning models on cancer patients who were infected with COVID-19, and further compared it to the general population. Cancer patients were reported to have worse COVID-19 outcomes, greater need for ventilator, and elevated mortality rates [37]. Specifically, we have demonstrated that models trained on a general population will not optimally perform when applying to cancer of phenotypically different radiographic patterns as driven by distinct underlying physiology. Systemic cancer treatment regimens can expose patients to an elevated infection risk and lead to worse COVID-19 outcomes [38]. Thus, our prognostic model can potentially shed light on informing the clinical cancer management. In this study, we have shown that the COVID-19 models trained separately on the cancer cohort outperformed those trained on the multicenter general population by a significant margin. This suggests that publicly available COVID-19 models which are developed and validated in the general population will not be optimally applied to cancer.

The superior performance of the proposed algorithms warrants its further validation for the vulnerable cancer population. If validated, the imaging-based analysis may add clinical value. For COVID-19+ cancers, our prognostic models can be used to identify high-risk patients, who may need urgent and intense medical care such as admission to hospital and ICU, and ventilation. In addition, the routine follow-up and screening CT scans can feed into our diagnostic model for accidental COVID-19 infection detection to prevent nosocomial transmission. In particular, our algorithm has potential clinical values in differentiating the COVID-19-induced pneumonia from the non-COVID-associated pneumonia/pneumonitis.

Currently, chest imaging of COVID-19 positive patient lungs exhibits variable infection patterns across different regions of the lung. Radiologists score their severity through assessing the percentage and distribution of distinct infected regions, such as GGO and consolidation. These standards are known to be qualitative, subjective, and not enough to quantify complex infection patterns. Thus, most existing AI approaches (including radiomics and deep learning) attempt to address these limitations through crafting robust computational pipelines to automate the radiologist’s annotation or semantic readings. For instance, Zhao et al. [39] developed an automatic technique to segment areas of pneumonia on CT scans and extract texture features for COVID-19 diagnosis. Bai et al. [40] developed a federated learning framework of deep learning to advance COVID-19 diagnosis on CT scans obtained from 22 hospitals, which is shown to be correlated with GGO, interlobular septal thickening and consolidation. However, historically these radiographic standards have been developed to assess pneumonia of other causes. A critical question is whether COVID-19 causes a distinct pattern of infection and whether the description of this pattern can be conveyed in the usual terminology used by radiologists. Here, we investigated this question by adapting previously validated habitat imaging pipeline [26,32,33], which has been used to identity intrinsic intra-tumoral subregions with differing imaging phenotypes (i.e., habitats).

This knowledge-discovery approach allows us to reveal more refined delineation of areas affected by pneumonia (i.e., COVID-habitats) by applying the unsupervised clustering algorithm on the CT scans. From the COVID-habitat map, it has demonstrated that these subregions not only overlapped with the areas of pneumonia as 3D annotated by the radiologist, but also divided these areas consistently into intrinsic subregions. Consequently, we have identified more refined six phenotypically distinct subregions within the lungs, which is in line with prior domain knowledge [36]. Of note, our habitat analysis leverages the unsupervised analysis to automate labeling these regions rather than leveraging the manual annotations from radiologists. Moreover, by correlating these habitat-derived features with radiologist’s semantic readings (see Appendix A), interestingly we observe that certain habitat features are significantly correlated to presence and quantity of GGO, quantity of GGO/consolidation, presence of bronchial wall thickening and pulmonary embolism. Of note, our proposed habitat approach does not require manual annotations from the radiologists, which are time consuming to obtain and often complicated with inter-observer variations. Moreover, we used unsupervised learning aims to explore data patterns which are inherently more robust than a supervised approach, less prone to model overfitting, and more likely to identify novel interactions between data that may not be readily apparent. As such, our approach can efficiently capture subtle patterns of infection and may be more robust on small and heterogeneous datasets as compared to deep learning and radiomics approaches. Given that our habitat model outperformed several existing radiomic [41,42,43,44] and deep learning approaches [15,45,46,47,48,49] (Appendix A), it is plausible that our approach can improve our understanding of the radiologic manifestation of SARS-CoV-2 pneumonia in cancer patients.

Our study has some limitations. First, we did not study whether our habitat imaging models could predict COVID-19 related deaths because we had limited follow up data. In the future, exploring how patterns of COVID-19 infection affect the overall survival and disease progression in cancer patients will help us better manage this high-risk population. Second, though we have enrolled multi-institutional datasets of different populations (general vs. cancer), these initial encouraging findings need to be further validated. Third, given that the COVID-habitat-based features are fundamentally driven by differing microenvironments in infected lung regions, future efforts are needed to explore the biological drivers of the subregions on CT scans.

In conclusion, we have developed and validated habitat imaging-based CT signatures to diagnose and predict the severity of COVID-19 in cancer and general population. These CT signatures have been developed and validated using data from multiple centers. Thus, the CT signatures showed the potential to help identify cancer patients who may benefit from urgent and intense care. These results warrant further verification in future prospective data to refine such findings and test clinical utility of these imaging biomarkers to manage cancer patients infected with COVID-19.

## Figures and Tables

**Figure 1 cancers-15-00275-f001:**
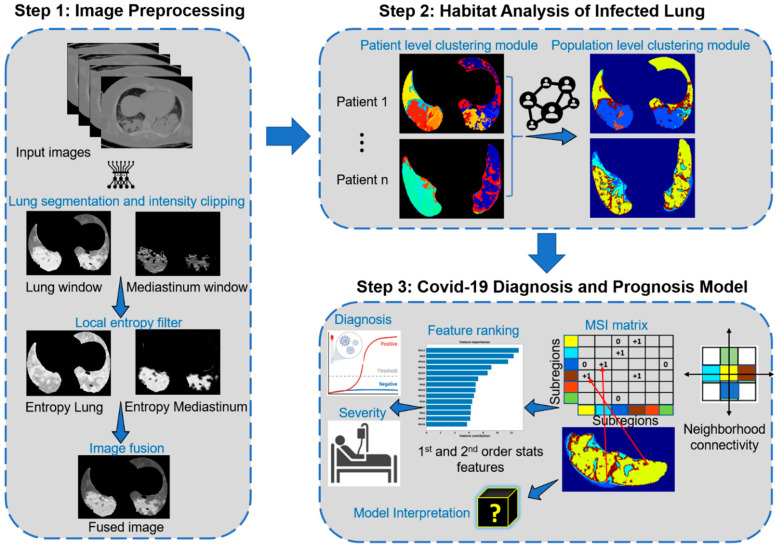
Workflow of the proposed approach.

**Figure 2 cancers-15-00275-f002:**
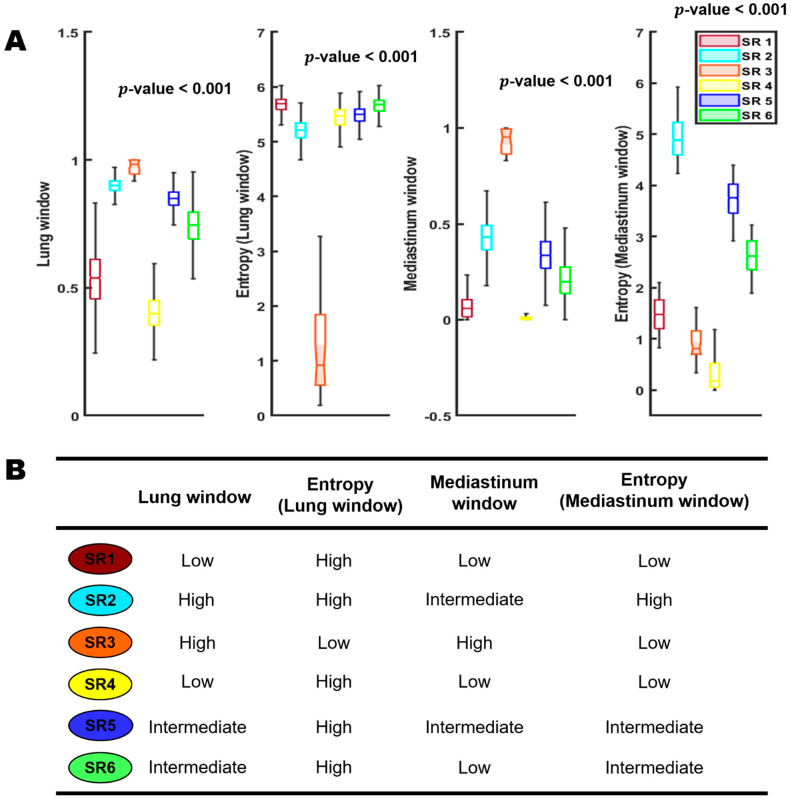
Habitat subregions distributions; SR: subregion. Row (**A**) is a boxplot showing the detailed distribution of the four input image channels across the six habitat regions. Row (**B**) represent the imaging interpretations of the six habitat regions.

**Figure 3 cancers-15-00275-f003:**
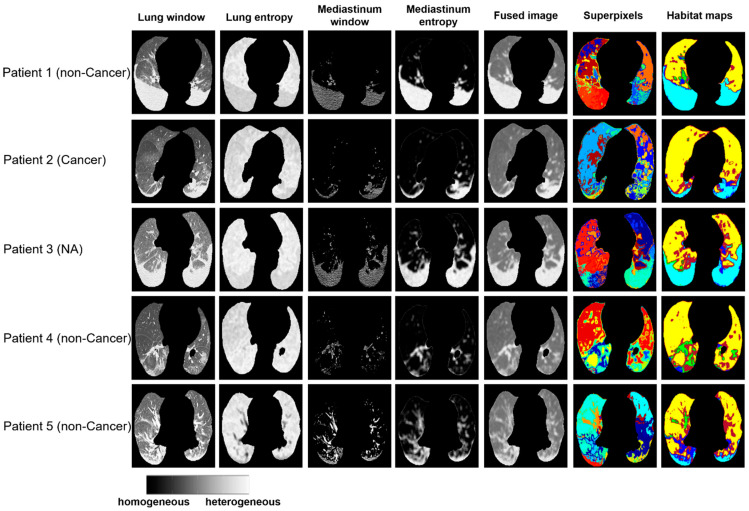
Examples of habitat maps of 5 cases from the Stony Brook dataset.

**Figure 4 cancers-15-00275-f004:**
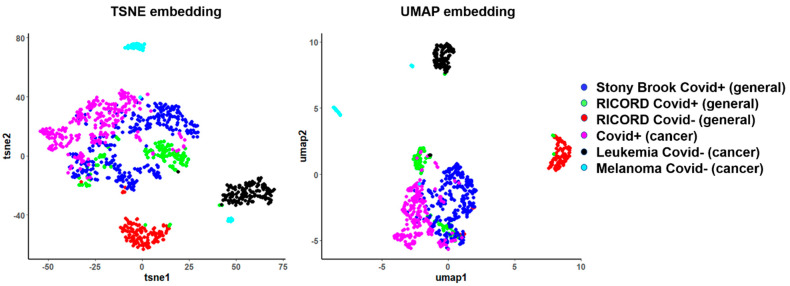
Two-dimensional embeddings of general population and cancer population datasets after habitat analysis.

**Figure 5 cancers-15-00275-f005:**
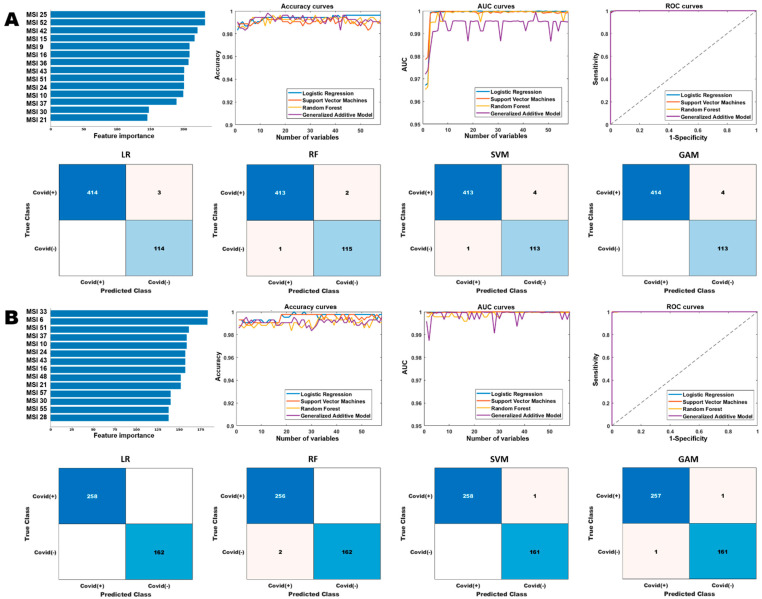
Performance comparison of the different diagnostic models based on the multiregional spatial interaction features. Row (**A**,**B**) show the performances of the different models on the general and cancer cohorts, respectively. The “Accuracy curves” and “AUC curves” shows the performance (classification accuracies and area under the curve) of the different classification models with respect to different number of selected features. The confusion matrices show the performance of the best models (in terms of the best selected features) and “ROC curves” shows the corresponding receiver operating characteristics curves of the best models.

**Figure 6 cancers-15-00275-f006:**
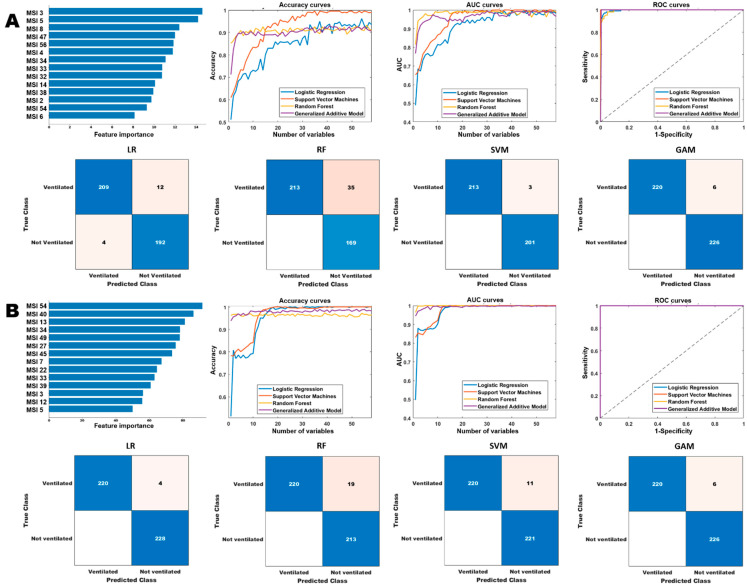
Comparison of the different models for ventilation prediction. Row (**A**,**B**) show the performances of the different models on the general and cancer cohorts, respectively. The “Accuracy curves” and “AUC curves” shows the performance (classification accuracies and area under the curve) of the different classification models with respect to different number of selected features. The confusion matrices show the performance of the best models (in terms of the best selected features) and “ROC curves” shows the corresponding receiver operating characteristics curves of the best models.

**Table 2 cancers-15-00275-t002:** Multiregional spatial interaction (MSI) features interpretation.

Feature Name	Feature Description
MSI 1–MSI 4	2nd order statistics features
MSI 5–MSI 10	absolute habitat subregions volume (SR1–SR6)
MSI 11–MSI 16	interaction (absolute) between habitat subregions and border
MSI 17–MSI 21	interaction (absolute) between SR1 and the remaining subregions, i.e., MSI 17 = SR1 ∩ SR2, MSI 18 = SR1 ∩ SR3, …, MSI 21 = SR1 ∩ SR6.
MSI 22–MSI 25	interaction (absolute) between SR2 and SR3, SR4, SR5 and SR6, i.e., MSI 22 = SR2 ∩ SR3, MSI 23 = SR2 ∩ SR4, MSI 24 = SR2 ∩ SR5, MSI 25 = SR2 ∩ SR6.
MSI 26–MSI 28	interaction (absolute) between SR3 and SR4, SR5 and SR6, i.e., MSI 26 = SR3 ∩ SR4, MSI 27 = SR3 ∩ SR5, MSI 28 = SR3 ∩ SR6.
MSI 29–MSI 30	interaction (absolute) between SR4 and SR5 and SR6, i.e., MSI 29 = SR4 ∩ SR5, MSI 30 = SR4 ∩ SR6.
MSI 31	interaction (absolute) between SR5 and SR6, i.e., MSI 30 = SR5 ∩ SR6.
MSI 32–MSI 37	percentage of habitat subregions volume (SR1–SR6)
MSI 38–MSI 43	normalized interaction (percentage) between habitat subregions and border
MSI 44–MSI 48	normalized interaction (percentage) between SR1 and the remaining subregions, i.e., MSI 44 = SR1 ∩ SR2, MSI 45 = SR1 ∩ SR3, …, MSI 48 = SR1 ∩ SR6.
MSI 49–MSI 52	normalized interaction (percentage) between SR2 and SR3, SR4, SR5 and SR6, i.e., MSI 49 = SR2 ∩ SR3, MSI 50 = SR2 ∩ SR4, MSI 51 = SR2 ∩ SR5, MSI 52 = SR2 ∩ SR6.
MSI 53–MSI 55	normalized interaction (percentage) between SR3 and SR4, SR5 and SR6, i.e., MSI 53 = SR3 ∩ SR4, MSI 54 = SR3 ∩ SR5, MSI 55 = SR3 ∩ SR6.
MSI 56–MSI 57	normalized interaction (percentage) between SR4 and SR5 and SR6, i.e., MSI 56 = SR4 ∩ SR5, MSI 57 = SR4 ∩ SR6.
MSI 58	normalized interaction (percentage) between SR5 and SR6, i.e., MSI 58 = SR5 ∩ SR6.

**Table 3 cancers-15-00275-t003:** Performance comparison of the different models for COVID-19 diagnosis on the general and cancer cohorts. Acc: accuracy; Sen: sensitivity; Spe: specificity; AUC: area under the receiver operating characteristic curve. LR: logistic regression, RF: random forest, SVM: support vector machine, GAM: generalized additive model.

Methods	Cohort							
	General				Cancer			
	Acc	Sen	Spe	AUC	Acc	Sen	Spe	AUC
LR	0.9944	1.0000	0.9928	0.9999	1.0000	1.0000	1.0000	1.0000
RF	0.9944	0.9914	0.9952	0.9998	0.9952	0.9878	1.0000	0.9998
SVM	0.9906	0.9912	0.9904	0.9998	0.9976	1.0000	0.9961	1.0000
GAM	0.9925	1.000	0.9904	0.9997	0.9952	0.9938	0.9961	1.0000

**Table 4 cancers-15-00275-t004:** Performance of the COVID-19 diagnostic models trained on the general cohort and applied on the cancer cohort.

Methods	All			
	Acc	Sen	Spe	AUC
LR	0.8738	0.9910	0.8317	0.9807
RF	0.8548	1.0000	0.8088	0.9178
SVM	0.8548	1.0000	0.8088	0.9586
GAM	0.9000	0.9918	0.8624	0.9202

**Table 5 cancers-15-00275-t005:** Performance comparison of the different models for ventilation prediction.

Methods	Cohort							
	General				Cancer			
	Acc	Sen	Spe	AUC	Acc	Sen	Spe	AUC
LR	0.9616	0.9796	0.9457	0.9961	0.9912	1.0000	0.9821	1.0000
RF	0.9161	1.0000	0.8589	0.9949	0.9580	1.0000	0.9205	1.0000
SVM	0.9928	1.0000	0.9861	1.0000	0.9757	1.0000	0.9524	1.0000
GAM	0.9161	1.0000	0.8589	0.9967	0.9867	1.0000	0.9735	1.0000

## Data Availability

The Stony Brook dataset can be found on The Cancer Imaging Archive (TCIA) at https://wiki.cancerimagingarchive.net/pages/viewpage.action?pageId=89096912#89096912bcab02c187174a288dbcbf95d26179e8 accessed on 1 November 2021. The RICORD dataset [30] can be found on https://wiki.cancerimagingarchive.net/pages/viewpage.action?pageId=80969742 accessed on 1 November 2021. Data from MD Anderson is currently not available at public repositories due to privacy and ethical issues but can be made available by the corresponding author upon reasonable request.

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
