# Peer review of "Habitat Imaging Biomarkers for Diagnosis and Prognosis in Cancer Patients Infected with COVID-19"

_cancers, 2022, doi:10.3390/cancers15010275_

Round 1
Reviewer 1 Report
This manuscript describes the development and validation of machine learning (ML) models for the prognosis/diagnosis of COVID-19 from habitat imaging biomarkers, as extracted from CT scans. A particular focus is the tailoring of the models towards cancer patients, which has been less-explored in the literature thus far. It was found that models developed on cancer patients performed better on the same cancer population (in cross-validation), as compared to models developed (as typically) on the general population. This has clear implications for precision/target medicine.
While promising, some issues might be clarified:
1. The habitat imaging pipeline process (as described in Section 2.4) might be illustrated with suitable figures for better understanding.
2. Table 1 in Section 2.2 states demographics for Stony Brook (N=275) and **(N=252). However, the text then mentions 531 CT scans from Stony Brook & RICORD for the general population, and 420 CT scans for the cancer population. The demographic numbers might thus be clarified.
3. In Section 2.3, it is stated that a 3D nnU-net was used for segmenting right and left lungs from the CT images. The training of the nnU-Net, or whether a trained model was used, might be clarified.
4. Related to the above, Section 3.1 states that six distinct habitats were consistently identified, in both the general and cancer cohorts. These six subregions (apparently SR1 to SR6 in Figure 2) might be illustrated in a suitable figure, and described.
5. It might be clarified as to whether there exist cases where fewer than (or more than) six subregions are segmented by the habitat imaging pipeline, or whether six habitats is guaranteed by prior definition of the number of clusters, at the global clustering step.
6. In Section 3.2, 58 features are stated to have been derived from the habitat maps, whereas it is stated in Section 2.6 that feature selection via feature importance was performed, later confirmed as the top 23/10/13/7 features for LR/SVM/RF/GAM respectively. The top features (and their importance) as selected for each ML model might thus be reported, possibly in supplementary material.
7. From Section 3.2, it is understood that separate models were trained and evaluated for the general and cancer cohort, i.e. one model trained/evaluated on general cohort data only, and another model trained/evaluated on cancer cohort data only. Natural questions then would be the performance of models trained on the full data (both general and cancer), and the performance of models trained on the cancer cohort, and applied to the general cohort (the converse of Table 4).
8. In addition, another natural approach would be the direct prediction of the outcome from the CT scan images, using suitable ML models such as VGGNet, ResNet, DenseNet etc. It might be clarified whether this approach was considered, as opposed to feature extraction via clustering and then classification according to those features.
9. Some minor grammatical/phrasing issues might be considered, e.g.:
(Lines 38/39) "build" -> "built"
(Line 365) "that explore the" -> "that explores the"
etc.
Author Response
Thank you very much for your kind and careful review of our manuscript.

Reviewer 2 Report
The paper presented an image representation method based on habitat imaging for Covid-19 detection and severity estimation compared with cancer-patient and general population.
Main issues from application perspective are (1) the data do not include community acquired pneumonia cases (neither in general population nor in cancer-cohort), (2) no comparison with any state of art method of deep learning on the same dataset used by the authors, (3) in prognosis, the input data used for the prediction are not clearly defined: how many days between the clinical outcome and the prediction input? If the ‘prediction’ is based on imaging date from the patients with the existing outcome of the severity (outpatient, inpatient, ventilation, ICU), it is not a proper claim as prognosis.
Habitat imaging, like radiomics, is used to represent tumor imaging property in imaging studies for cancers. This paper should compare with the state of art radiomics based method for the proposed tasks. However it is only compared the pipeline with different ML methods (LR, SVM, GAM, RF), all based on the same habitat imaging features.
Other questions the authors could address:
1, Is control group too limited to the cancer patients without Covid-19? How about other community acquired pneumonia (CAP) cases?
2, What is the Covid-19 positive defined, as PCR positive only or the data are actually with pneumonia seen in the CT? It is well known now PCR+ may have no pneumonia, which will then not show any lung infections.
Typo: ceells (cells)
For the cancer population, if there are other CAP pneumonia, will the method work to detect the Covid-19 infection? If not, the paper should focus on the severity classification (or as claimed prognosis of outcome).
In Table I, it is not very clear. It should be written to make the figures clear. The figures (275+252=527) can not be synced with the text description. What is ** with N=252?
What are the N=275 and N=252 subjects? Which one is general population, cancer population, Covid-19 positive (by PCR+?), Covid-19 negative?
In the text write-up, there are 531 CTs from general population, 385 with Covid-19 with PCR+.
Is the Hospitalization status are the ‘severity’ or ‘prognosis’ target? As mentioned before, what are the difference between the imaging date and the date with the confirmed hospitalization status?
What is the relationship between the CT and the PCR (time difference)?
Line 146, what is the way to obtain the ‘fused image’? by simply adding?
Line 187, how to define / compute co-occurring pairs?
Lin 225, explanation on ‘association’ using the volcano plot is missing
In Table 2, what is tumor subregion volume? Do you do tumor segmentation for the lung cancer?
Line 274, non-covid-19-pneumoina still cancer caused? Any CAP pneumonia? The main question is, if the proposed method can differentiate Covid-19 and other lung infection pneumonia.
In Figure 4, Should include other pneumonia (community acquired pneumonia) cases to show the capability to separate them in order to detect Covid-19.
Lin 279: Does ”detect Covid-19 infection” mean pneumonia can be seen? Although the paper defined the infection as PCR positive.
Lin 299-300, Comparison is not valid. The authors should compare with those CT based deep learning method and apply to your data (same data). Otherwise the comparison is not meaningful to claim best performance.
How about using deep-feature (learnt by a deep learning network) and applying machine learning method (LR, SVM, GAM etc)?
Table 4: why only compare between conventional machine learning methods? No advanced deep learning method (CNN based, e.g. ResNet, DenseNet, Vision Transformer) tested on the data?
Line 317-319: not valid / clear without the time info.
Table 5, That AUC=1 for Cancer group means the data is too easy to classify? How about the result of using any state-of-art deep learning methods on the same data?
If AUC=1, should Sen and Spe both equal to 1 as well?
Lin 416-417, not clear how to interpret the correlation from Fig.5.
Author Response

(The authors gave the same response as above.)

Round 2
Reviewer 1 Report
We thank the authors for addressing our previous concerns. Some further minor clarifications might be considered:
1. The training details for the nnU-net used for segmenting left and right lungs, might be described. In particular, what training data was used?
2. The new results for deep learning (Section 3.6) might be presented in the same format as the main results (e.g. Tables 3 to 5, Figure 5 and 6), for easy comparison.
Author Response
We have included tables in the supplementary materials showing the performance of the different models when fed in the deep features. Thank you very much for your suggestions to help improve our work.
